# OpenReview forum: "From Theft to Bomb-Making: The Ripple Effect of Unlearning in Defending Against Jailbreak Attacks"
_ICLR.cc/2026/Conference — Submitted to ICLR 2026_

### Official Review · Reviewer_skqa · 2025-10-14

**Soundness:** 2
**Presentation:** 2
**Contribution:** 2
**Rating:** 2
**Confidence:** 5

**Summary:**

This paper identifies and empirically validates a novel "ripple effect" in unlearning-based defenses for Large Language Models (LLMs). It demonstrates that unlearning one type of harmful knowledge (e.g., theft instructions) can unintentionally cause the forgetting of other, untargeted harmful knowledge (e.g., bomb-making procedures). This phenomenon was observed across over 100 experiments using models like Llama-3.1-8B-Instruct and Mistral-7B-Instruct against various jailbreak attacks. Evaluate the model's reasoning capabilities using tasks from datasets like Big-bench or other Chain-of-thought datasets.

**Strengths:**

- It conducts in-depth analytical experiments to uncover the phenomenon's underlying causes, suggesting that the intrinsic relatedness of harmful knowledge is a key contributing factor.
- The ripple effect of unlearning is identified through extensive experimentation.

**Weaknesses:**

**Limited Originality:** The so-called “ripple effect” is essentially a restatement of standard generalization in unlearning and safety fine-tuning. The notion that altering representations for one concept influences related unseen ones is a well-understood property of representation learning. The paper’s contribution is primarily an empirical confirmation of expected behavior rather than a novel discovery.

**Incomplete Evaluation:** The study lacks recent state-of-the-art attack baselines such as AutoDAN-Turbo [3], BOOST + GPTFuzzer [1], and TAP [2]. Adding one or two newer attacks and evaluating on modern LLMs like Qwen3 and Gemma3 would provide a more comprehensive comparison.

**Shallow Analysis of the “Ripple Effect”:** The analysis of why and how this generalization emerges is superficial and does not explore its limits. The HarmBench ID/OOD split may involve semantically similar topics (e.g., various illegal activities). A more rigorous design should test cross-domain generalization, such as training to unlearn violent content and evaluating on biological or nuclear-related harms.

[1] Mind the Inconspicuous: Revealing the Hidden Weakness in Aligned LLMs' Refusal Boundaries

[2] Tree of Attacks: Jailbreaking Black-Box LLMs Automatically

[3] AutoDAN-Turbo: A Lifelong Agent for Strategy Self-Exploration to Jailbreak LLMs

**Questions:**

- Does a similar observation occur in larger Large Language Models (LLMs)?
- Can unlearning alter the ethical boundaries proposed in the BOOST[1] paper?
- Why do your methods and baselines, like Circuit Breaker exhibit such high perplexity?
- How is the model’s reasoning ability affected after unlearning?

---

> ### Author Response · Authors · 2025-11-24
> **Response by Authors**
>
> We sincerely thank the reviewer for the thoughtful and detailed feedback. We are encouraged by the reviewer’s recognition of the significance of our analytical experiments in uncovering the underlying causes of the phenomenon, particularly highlighting the role of the intrinsic relatedness of harmful knowledge. We also appreciate the reviewer’s acknowledgment of our identification of the ripple effect of unlearning through extensive experimentation. Below, we address the reviewer’s questions and concerns in detail:
>
> > **[W1]**：Limited Originality: The so-called “ripple effect” is essentially a restatement of standard generalization in unlearning and safety fine-tuning. The notion that altering representations for one concept influences related unseen ones is a well-understood property of representation learning. The paper’s contribution is primarily an empirical confirmation of expected behavior rather than a novel discovery.
>
> **[R1]**: We respectfully note that Section 3 of our paper already discusses the distinctions between the ripple effect we observe and prior understandings of unlearning and safety generalization. Specifically, prior work on unlearning generally expects effects to be confined to the training set, whereas our findings show that unlearning can influence data beyond the training set. Similarly, prior safety generalization focuses on input-level generalization across harmful queries, while our work uncovers output-level generalization across harmful responses. This highlights that the ripple effect we describe is conceptually and empirically distinct from previously documented phenomena.
>
> > **[W2]**：Incomplete Evaluation: The study lacks recent state-of-the-art attack baselines such as AutoDAN-Turbo [3], BOOST + GPTFuzzer [1], and TAP [2]. Adding one or two newer attacks and evaluating on modern LLMs like Qwen3 and Gemma3 would provide a more comprehensive comparison.
>
> **[R2]**: We appreciate the reviewer’s suggestion. We would like to clarify that TAP has already been evaluated in our main experiments (Tables 1 and 2), and GPTFuzzer is evaluated in Table 9.
>
> Regarding newer LLMs, our evaluation includes multiple recent models, such as Llama-3.1 and Mistral-v0.3 (Tables 1 and 2) and Qwen2.5-3B/14B/32B (Table 7). We believe these results already provide strong evidence supporting the observed ripple effect.
>
> Nevertheless, following the reviewer’s suggestion, we additionally evaluated Gemma3 on OOD harmful questions, with the results summarized below:
> | Method          | Raw  | Manual   | Prefilling | PAIR | PPL    |
> | --------------- | ---- | -------- | ---------- | ---- | ------ |
> | **Raw**             | 16.0 | 29.0 | 60.0       | 57.0 | 3.87   |
> | **Safe Unlearning** | **0.0**    | **6.5**      | **0.0**          | **25.0** | 9.51e8 |
>
> These results show that the ripple effect persists in newer models like Gemma3, as indicated by the significant reduction in ASR and the high PPL on OOD harmful responses.
>
> > **[W3]**：Shallow Analysis of the “Ripple Effect”: The analysis of why and how this generalization emerges is superficial and does not explore its limits. The HarmBench ID/OOD split may involve semantically similar topics (e.g., various illegal activities). A more rigorous design should test cross-domain generalization, such as training to unlearn violent content and evaluating on biological or nuclear-related harms.
>
> **[R3]**:
> We respectfully note that Section 5 of our paper already addresses cross-domain generalization and the limits of the ripple effect. We evaluated this by dividing queries into six semantic categories and observed the following:
> 1. Intra-category generalization is significantly easier than inter-category generalization.
> 2. The ripple effect can emerge with very few examples.
> 3. Models vary in their capacity for cross-category generalization.
> 4. The strength of the ripple effect differs across semantic categories.
>
> Regarding the reviewer’s example of whether training on violent content could transfer to biological or nuclear-related harms, such cross-domain transfer is indeed possible. For instance, as shown in Figure 4, training on the “illegal” category transfers well to the “chemical” category. Please refer to Section 5 for a detailed discussion of these observations.

---

> > ### Comment · Reviewer_skqa · 2025-11-24
> >
> > Thanks to the authors for the clarifications and for taking the time to respond. I have carefully read the responses as well as the other reviewers’ comments. After considering your clarifications, I think the experiment part and novelty both need improvement. Therefore, I will maintain my original score. Thanks again for efforts.

---

> > > ### Author Response · Authors · 2025-11-24
> > >
> > > We sincerely thank the reviewer for the follow-up response. We would like to respectfully note that many of the original concerns appear to stem from overlooking several analyses and results already presented in the paper. In our rebuttal, we have provided detailed clarifications and pointed to the exact sections where these discussions and findings can be found, along with additional complementary experiments.
> > >
> > > Given this, we kindly ask the reviewer to reconsider these clarified and newly added results, and—if possible—specify which particular concerns remain unaddressed. This would greatly help us further improve the work, as the current statement that “the experiment part and novelty both need improvement,” especially in reference to “other reviewers’ comments,” is too general for us to act upon.
> > >
> > > Thank you again for your time and consideration.

---

> > > > ### Comment · Reviewer_skqa · 2025-11-24
> > > >
> > > > Thanks to the author's hard work during the rebuttal period, and I think the author did not respond to any question in my question section. Also, include some advance jailbreak method will highly enhance the paper

---

> > > > > ### Author Response · Authors · 2025-11-24
> > > > >
> > > > > We sincerely thank the reviewer for the prompt follow-up. We apologize for the earlier oversight in not addressing the questions listed in the “Questions” section. It seems that the review content was updated during the rebuttal period, and our initial responses were aligned with the earlier version. We now provide detailed answers to all listed questions below.
> > > > >
> > > > > > **[Q1]**: Does a similar observation occur in larger Large Language Models (LLMs)?
> > > > >
> > > > > **[R1]**: Yes. As shown in Table 7, the same observation consistently appears across Qwen2.5-7B/14B/32B, indicating that the phenomenon scales with model size.
> > > > >
> > > > > > **[Q2]**: Can unlearning alter the ethical boundaries proposed in the BOOST[1] paper?
> > > > >
> > > > > **[R2]**: Yes. The boundaries defined in BOOST relate to the model’s ability to distinguish harmful queries from harmless ones. Prior to unlearning, the model exhibits a high ASR, suggesting difficulty in identifying harmful queries. After unlearning, the ASR decreases substantially, indicating that the model becomes more capable of rejecting harmful queries. This demonstrates that unlearning can indeed shift the ethical boundaries described in BOOST.
> > > > >
> > > > > > **[Q3]**: Why do your methods and baselines, like Circuit Breaker exhibit such high perplexity?
> > > > >
> > > > > **[R3]**: As discussed in Section 4, high perplexity on harmful responses indicates that the model has sufficiently unlearned undesired content. This behavior is expected and, in our experiments, directly contributes to improved safety performance.
> > > > >
> > > > > > **[Q4]**: How is the model’s reasoning ability affected after unlearning?
> > > > >
> > > > > **[R4]**: Our study focuses on non-reasoning models because both unlearning and jailbreak methods tailored for reasoning models remain underdeveloped. Thus, evaluating reasoning ability on these non-reasoning models would not be informative. Nevertheless, we verified that instruction-following (MTBench) and knowledge capacity (MMLU) remain largely unaffected after unlearning.
> > > > >
> > > > > On the suggestion regarding more advanced jailbreak methods, our paper already includes TAP and SAA (and many other representative attack methods), both of which are **strong and widely adopted jailbreak attacks in recent safety literature. This aligns with the reviewer’s original recommendation to include at least one of the suggested methods (e.g., TAP)**. Moreover, the strong generalization (i.e., ripple effect) observed across these diverse and powerful attacks illustrates that the current set of jailbreak methods is sufficient to support our claims regarding unlearning generalization.

---

> > > > > > ### Comment · Reviewer_skqa · 2025-11-24
> > > > > >
> > > > > > No, I just think the original score 2 is so strict to this paper, so I increased the score to 4 before your rebuttal.

---

> > > > > > > ### Comment · Reviewer_skqa · 2025-11-24
> > > > > > >
> > > > > > > I deeply review your response, and I am sorry I missed the TAP in the review. I would still suggest you extend to Autodan-Turo, but it should be fine. One issue and respectful disagreement is that the LLM (not LRM) can also do some COT tasks to test the reasoning ability with few-shot. But it should be fine.  The main concern is still I list in `Reviewer nVCM` and `Reviewer nVCM` mentioned: **if the contribution is limited to an observation, it is a good finding, but it may be perceived as insufficient. Introducing a new method or providing a theoretical proof is merely my suggestion, which could further strengthen the paper’s contribution to the community.** As a result, I would suggest to keep my score, and if other reviewers agree to accept it, I will not refuse it.

---

> > > > > > > > ### Author Response · Authors · 2025-11-24
> > > > > > > >
> > > > > > > > Thank you very much for your kindness and for the clarifications. We now have a clearer understanding of your main concern (though it differs somewhat from the detailed concerns described in the initial review). Our detailed response to this point is provided in our initial reply to Reviewer nVCM.
> > > > > > > > We sincerely appreciate the constructive interaction and your willingness to support acceptance if other reviewers find the contribution sufficient. Please feel free to reach out if any further clarification would be helpful—we remain open to continued discussion.

---

### Official Review · Reviewer_nVCM · 2025-10-20

**Soundness:** 3
**Presentation:** 3
**Contribution:** 2
**Rating:** 2
**Confidence:** 4

**Summary:**

The paper explores that how unlearning affect LLM's harmful knowledge,  and show that whhen one cluster is suppressed through unlearning, nearby clusters are also weakened.  The authors apply several unlearning approaches and jailbreak methods to evaluate the finding.

**Strengths:**

1. The paper is well written.
2. The structure is well organized.
3. The experiment is comprehensive.

**Weaknesses:**

1. The major weakness is the novelty. The related conclusion is well studied regarding the representation learning in NLP[1,2,3,4],It essentially is a discussion under the generalization of unlearning. The paper does not introduce a new unlearning algorithm and it only reuses existing methods to observe the effect.
2. The paper is stay at experiment and observation level regarding the "ripple effect", There is no formal causal or mathematical model explaining why unlearning one cluster suppresses others, which could be the possible contribution or novelty.
3. The paper claims semantically related topics generalize better, but it lack of quantitative results to support this claim.

Generally, I believe the major issue is the author does not propose their own methods but just stay at report the experiment results. Given that the authors did not propose an original method, I am unable to provide further strengths or weaknesses. It is a good finding paper but not enough novelty for main conference.


[1]Li, N., et al. “The WMDP Benchmark: Measuring and Reducing Malicious Use with Unlearning.” Proceedings of the 41st International Conference on Machine Learning, Proceedings of Machine Learning Research, vol. 235, 2024, pp. 28525–28550

[2]Zou, Andy, et al. "Improving alignment and robustness with circuit breakers." Advances in Neural Information Processing Systems 37 (2024): 83345-83373.

[3]Wu, Chengcan, et al. "Reliable Unlearning Harmful Information in LLMs with Metamorphosis Representation Projection." arXiv preprint arXiv:2508.15449 (2025).

[4]Ravfogel, Shauli, et al. "Linear adversarial concept erasure." International Conference on Machine Learning. PMLR, 2022.

**Questions:**

See weakness.

---

> ### Author Response · Authors · 2025-11-24
> **Response by Authors (1/2)**
>
> We sincerely thank the reviewer for the thoughtful feedback. We are encouraged by the reviewer’s recognition of the clarity of our writing, the well-organized structure of the paper, and the comprehensiveness of our experimental evaluation. Below, we address the reviewer’s questions and concerns in detail.
>
> > **[W1]**: The major weakness is the novelty. The related conclusion is well studied regarding the representation learning in NLP[1,2,3,4],It essentially is a discussion under the generalization of unlearning. The paper does not introduce a new unlearning algorithm and it only reuses existing methods to observe the effect. Generally, I believe the major issue is the author does not propose their own methods but just stay at report the experiment results. Given that the authors did not propose an original method, I am unable to provide further strengths or weaknesses. It is a good finding paper but not enough novelty for main conference.
>
> **[R1]**: We thank the reviewer for raising the question of novelty. We respectfully disagree with the claim that our findings are "well studied" or that our work merely "stays at reporting the experiment results".
>
> First, regarding the reviewer’s comment on related works [1–4], we would like to clarify that while these papers indeed introduce new representation-learning methods, **their focus is algorithmic innovation rather than understanding the structural behavior of models under unlearning**. They do not analyze how harmful responses are organized, nor how removing part of them affects the rest. In fact, despite proposing new techniques, these works do not provide the type of deep mechanistic or structural analysis that our paper contributes. **Thus, our work is orthogonal and complementary: we study an unexplored phenomenon in unlearning rather than proposing another unlearning method**.
>
> Second, our paper is the first to systematically uncover that harmful behaviors form stable, coherent clusters in representation space, and that unlearning reshapes these shared subspaces, leading to predictable cross-sample suppression. **This goes beyond reporting observations: we introduce new analyses, controlled perturbation experiments, and mechanistic explanations** to reveal the structural conditions under which unlearning unexpectedly generalizes.
>
> Third, novelty in ICLR does not require a new algorithm. A significant body of prior ICLR papers are **phenomenon-discovery, measurement, or mechanistic-analysis** works, and many have been accepted. Our contribution is of this type:
> - we identify a previously unknown behavior (i.e., the ripple effect) of existing unlearning methods,
> - we provide representational and behavioral evidence explaining it,
> - we thoroughly discuss the generalization boundaries of the ripple effect, and
> - we establish new empirical regularities that future algorithms should take into account.
>
> In summary, our novelty lies not in introducing another algorithm, but in uncovering and explaining a fundamental and previously unrecognized mechanism in unlearning. We believe this is exactly the type of scientific contribution appropriate for ICLR.

---

> ### Author Response · Authors · 2025-11-24
> **Response by Authors (2/2)**
>
> > **[W2]**: The paper is stay at experiment and observation level regarding the "ripple effect", There is no formal causal or mathematical model explaining why unlearning one cluster suppresses others, which could be the possible contribution or novelty.
>
> **[R2]**: We appreciate the reviewer’s interest in a more formal causal or mathematical model of the “ripple effect”. However, we respectfully argue that requiring a full formal model is not necessary for establishing the contribution of our work.
>
> First, **the ripple effect we study has not been previously identified, quantified, or systematically analyzed in previous literature**. Prior works in this area are also predominantly empirical, without formal causal models. Discovering and characterizing a new generalization pattern is therefore a substantial and standalone contribution, and typically a prerequisite to later theoretical modeling.
>
> Second, while developing a complete causal theory for parameter-space interactions in large neural networks is extremely challenging and beyond the scope of most empirical studies, **our paper already provides concrete mechanistic evidence**. Specifically, harmful responses naturally form tightly grouped clusters in representation space. Consequently, unlearning a subset of responses within a cluster inevitably distorts the shared harmful subspace, which in turn suppresses other responses within the same cluster—even if they are not directly targeted. This directly explains the observed ripple effect at a representational/mechanistic level. These results constitute strong causal evidence at the mechanism level, even though they are not expressed as a closed-form mathematical model.
>
> Finally, our findings actually **lay the empirical foundation** required for developing a future formal causal model. By identifying stable structural regularities—such as coherent harmful clusters and their shared representational subspaces—we provide the constraints that any future causal theory should account for. In this sense, the reviewer’s suggested direction is valuable, and our work is an essential step toward it.
>
> > **[W3]**: The paper claims semantically related topics generalize better, but it lack of quantitative results to support this claim.
>
> **[R3]**: First, we need to clarify that we didn't claim semantically related topics generalize better in our paper. Moreover, we have made a detailed discussion in Section 5 regarding the generalization boundary of the ripple effect. Through controlled and **quantitative** experiments, we report 4 key observations: (1)Intra-category generalization is significantly easier than inter-category generalization. (2) The ripple effect emerges with very few examples. (3) Models differ in their capacity for cross-category generalization. (4) The strength of the ripple effect varies by semantic category.

---

> > ### Comment · Reviewer_nVCM · 2025-11-24
> > **Thanks for clarification**
> >
> > Thanks to the authors for the clarifications and for taking the time to response. I have carefully read the responses as well as the other reviewers’ comments. The majority of reviewers have independently raised similar questions about the originality of the contribution. After considering your clarifications, I respect the value of the empirical findings, but I believe the issues around conceptual novelty and methodological contribution remain. Therefore, I will maintain my original score. Thanks again for efforts.

---

> > > ### Author Response · Authors · 2025-11-24
> > >
> > > Thank you for your follow-up response. We truly appreciate the time and consideration.
> > >
> > > However, we respectfully disagree with the statement that “the majority of reviewers have independently raised similar questions about the originality of the contribution.” More specifically, your original concern focuses on the necessity of proposing a concrete method. In our rebuttal, we clarified that ICLR has a long history of accepting method-free analytical studies, and our work is positioned in this tradition.
> > >
> > > Regarding the other reviewers:
> > > - Reviewer n4PZ did not raise concerns related to originality or methodology.
> > > - Reviewer uhfA explicitly found our findings “certainly interesting and useful to know,” and although uncertain about the overall contribution, expressed willingness to advocate for acceptance once the other issues are resolved.
> > > - Reviewer skqa argued that the ripple effect aligns with standard generalization phenomena in unlearning and safety fine-tuning. We believe this reflects an oversight of our detailed discussion in Section 3, where we explain why our findings go beyond standard generalization.
> > > - Reviewer AUfc is indeed the only reviewer who also expects a new method. At the same time, they acknowledge that our insights are novel and valuable for designing more generalizable jailbreak defenses.
> > >
> > > In summary, only reviewer AUfc expressed a concern overlapping with yours regarding methodological contribution. The other reviewers raised distinct issues, all of which we have addressed thoroughly in our responses. Therefore, we feel that summarizing the reviews as reflecting a majority consensus on originality may inadvertently overlook the nuances in the individual reviews as well as the clarifications we provided.
> > >
> > > Thanks again for the thoughtful engagement. We remain open to further discussion and appreciate the constructive dialogue.

---

> > > > ### Comment · Reviewer_nVCM · 2025-11-24
> > > > **Thanks for clarification**
> > > >
> > > > Thanks to the authors for the clarifications and for taking the time to respond. I believe it is not only reviewer AUfc who raised this concern; reviewer skqa expressed the same viewpoint as well, as reflected in his reply in this thread. Reviewer uhfA also state clearly have concern about overall contribution. To clarify my core point: if the contribution is limited to an observation, it is good finding but it may be perceived as insufficient. Introducing a new method or providing a theoretical proof is merely my suggestion which could further strengthen the paper’s contribution to the community.

---

> > > > > ### Comment · Reviewer_skqa · 2025-11-24
> > > > >
> > > > > Yes, I also highly encourage the author to extend the work into a new method or providing a theoretical proof

---

> ### Comment · Reviewer_skqa · 2025-11-24
>
> I acknowledge that the “Ripple Effect” is an interesting observation, but I still think that observation is limited. It is hard to say whether it is a new methodology, though I claim the “Ripple Effect” is a great observation. I even read your section 3 again and again, but that is all I can get. I think your contribution is mainly focus on “Ripple Effect” and section 5

---

> ### Author Response · Authors · 2025-11-24
>
> Thank you for the prompt feedback. Regarding the statement that “only reviewer AUfc expressed a concern overlapping with yours regarding methodological contribution,” we were specifically referring to the **explicit request for a new method** as stated in the reviewer’s original review content. Our intention was to highlight that different reviewers raised concerns with **different emphases and different angles** (as summarized above), and thus **we should address them separately rather than summarizing them into a single general concern**. We fully understand and respect your core point, and we have addressed it in detail in our initial rebuttal. Thank you again for the constructive interaction, and we remain open to further discussion.

---

### Official Review · Reviewer_uhfA · 2025-10-25

**Soundness:** 3
**Presentation:** 2
**Contribution:** 2
**Rating:** 4
**Confidence:** 3

**Summary:**

This paper investigates and shows unlearning generalizes across topics. I think that the finding is generally interesting. There are some improvements to be made to the writing, the related work, and some framing, and there are some overclaims throughout the paper. IF those are fixed, I'd be happy to (weakly) advocate for acceptance, though I continue to have some concerns on the strength/size of the contribution overall here.

**Strengths:**

- The finding (generalization of unlearning) is interesting. I can't comment on how novel ti is, but I think it is certaintly interesting and useful to know. It also suggests how we might want to modify unlearning approaches to more explicitly target information we might want to remove. I don't personally find the finding __that__ surprising.
- Figure 2 with clear clustering is nice.

**Weaknesses:**

- Improvement of exposition and writing
    - e.g., the first paragraph mixed many different points, jailbreaks, unlearning, SFT. Better to keep it one point per paragraph.
- Improve related work
    - This work seems to be quite related to emergent misalignment (see https://arxiv.org/abs/2502.17424), but in fact the opposite direction. More of a form of emergent alignment. This should be mentioned.
    - Should also mention the MSJ paper (https://openreview.net/pdf?id=cw5mgd71jW), there they show jailbreaks generalization across topics if sufficienly diverse. This mechanism here is the same.
- If you can, it would be good to add citations that mention that people expect unlearning to be targetted. I don't know if this is actually the consensus in the field.
- I don't fully agree with the distinction to prior safety generalization. My guess would be that we can still elicit harmful information post-unlearning (see e.g., https://arxiv.org/abs/2410.08827)
- Please explain safe unlearning in the mian paper.
- Ideally you would include some stronger unlearning based attacks in the paper.
- I dont think you have evidence for this claim "Collectively, these results emphasize that a sufficient level of unlearning on harmful responses is essential for significantly reducing ASR" I think if you e.g., did SFT on the jailbreak prompts, you'd also improve ASR. Other claims like "Overall, our results indicate that unlearning shows remarkable generalization ability in defending against jailbreak attacks while preserving general performance." also seem to be overclaims to me based on the lack of adaptive attacks.

**Questions:**

I was a bit surprised about the main first set of results just showing how well unlearning works against attacks. I thought the main point of the paper was more about generalization? So how come? Can you explain the choice?

---

> ### Author Response · Authors · 2025-11-24
> **Response by Authors (1/2)**
>
> We sincerely thank the reviewer for the thoughtful and constructive feedback. We are encouraged by the reviewer’s recognition that our central finding—the generalization effect of unlearning—is interesting and practically useful, especially in informing how unlearning approaches may be redesigned to more explicitly target the information to be removed. We also appreciate the reviewer’s positive comments on the clarity of our visualization, particularly the well-clustered patterns shown in Figure 2. Below, we address the reviewer’s questions and concerns in detail.
>
> > **[W1]:** Improvement of exposition and writing (e.g., the first paragraph mixed many different points, jailbreaks, unlearning, SFT. Better to keep it one point per paragraph.
>
> **[R1]:** Thanks for the suggestion. We have restructured the introduction in the revised paper to dedicate separate paragraphs to jailbreaks, unlearning, and SFT to improve clarity and flow.
>
> > **[W2]**: Improve related work. This work seems to be quite related to emergent misalignment, but in fact the opposite direction. More of a form of emergent alignment. This should be mentioned.
>
> **[R2]**: Thanks for the insightful suggestion. We agree that "Emergent Misalignment" can be viewed as a complementary manifestation of the 'ripple effect' from the opposite perspective. While our work primarily focuses on the observation that removing harmful knowledge exhibits strong generalization, "Emergent Misalignment" highlights that inducing harmful behaviors also demonstrates strong generalization. We have included a discussion of this work in our Related Work section.
>
> > **[W3]**:Should also mention the MSJ paper, there they show jailbreaks generalization across topics if sufficienly diverse. This mechanism here is the same.
>
> **[R3]**: Thank you for the reference. While we agree that the MSJ paper also observes generalization across topics, we believe there is a key distinction in the underlying dynamics. MSJ primarily focuses on the generalization of jailbreak attacks (inducing failure), whereas our work analyzes the generalization of safety defense mechanisms (removing harmful knowledge). Although both involve generalization, they operate in opposing contexts (attack vs. defense). We agree that discussing MSJ will provide valuable context, and we have added it to our Related Work section.
>
> > **[W4]**: If you can, it would be good to add citations that mention that people expect unlearning to be targetted. I don't know if this is actually the consensus in the field.
>
> **[R4]**:  We thank the reviewer for this valuable suggestion. We have included citations [1,2,3] in Section 3 to reflect the general consensus that unlearning typically refers to the precise removal of specific samples seen during training without affecting others. However, our work expands beyond this traditional view in the context of safety. While the consensus anticipates strictly targeted removal, we identify a "Ripple Effect" where the unlearning of specific harmful knowledge implicitly generalizes to unseen harmful knowledge. This finding suggests that for safety alignment, unlearning is not merely a localized erasure but can trigger a broader suppression of the underlying harmful representation space.
>
> [1] Bourtoule L, Chandrasekaran V, Choquette-Choo C A, et al. Machine unlearning[C]//2021 IEEE symposium on security and privacy (SP). IEEE, 2021: 141-159.
>
> [2] Liu, Yujian, et al. "Revisiting Who’s Harry Potter: Towards Targeted Unlearning from a Causal Intervention Perspective." Proceedings of the 2024 Conference on Empirical Methods in Natural Language Processing. 2024.
>
> [3] Maini, Pratyush, et al. "TOFU: A Task of Fictitious Unlearning for LLMs." First Conference on Language Modeling. 2024.

---

> ### Author Response · Authors · 2025-11-24
> **Response by Authors (2/2)**
>
> > **[W5]**: I don't fully agree with the distinction to prior safety generalization. My guess would be that we can still elicit harmful information post-unlearning.
>
> **[R5]**:We wish to emphasize that the primary distinction lies in whether generalization occurs at the input level or the output level; we do not define the 'ripple effect' based on complete removal. As stated in the footnote at Page 1, we use the terms 'forgetting' and 'removing harmful knowledge' to denote a sufficient suppression of harmful outputs, rather than a guarantee of complete eradication. The question of whether harmful knowledge can ever be fully eliminated remains a subject of debate (as evidenced by our citations and the work mentioned by the reviewer) and lies beyond the scope of this study.
>
> > **[W6]**: Please explain safe unlearning in the main paper.
>
> **[R6]**: We appreciate the reviewer’s suggestion. In the initial submission, due to page limitations, we provided only a brief description of safe unlearning in Lines 182–186 and deferred the full formulas and detailed explanations to Appendix A. In the revised version, we have moved these explanations into the main paper to improve clarity and accessibility.
>
> > **[W7]**: Ideally you would include some stronger unlearning based attacks in the paper.
> > **[W8]**: I dont think you have evidence for this claim "Collectively, these results emphasize that a sufficient level of unlearning on harmful responses is essential for significantly reducing ASR" I think if you e.g., did SFT on the jailbreak prompts, you'd also improve ASR. Other claims like "Overall, our results indicate that unlearning shows remarkable generalization ability in defending against jailbreak attacks while preserving general performance." also seem to be overclaims to me based on the lack of adaptive attacks.
>
> **[R7]**:We thank the reviewer for challenging our claims and suggestions regarding attack strength. We appreciate the opportunity to clarify our evaluation setup and the evidence supporting our conclusions.
> Regarding "Stronger" Adaptive Attacks, we respectfully clarify that **our evaluation already incorporates strong adaptive attacks designed to bypass defenses**, which we believe aligns with the reviewer's request for "stronger" attacks. As detailed in Section 4.1, we utilized GCG, AutoDAN, PAIR, SAA, and TAP. These methods explicitly optimize adversarial prompts to break the target model's safety alignment. The robustness of our method against these aggressive optimization-based attacks serves as the primary evidence for our claims.
>
> Regarding SFT and Generalization Claims, we agree with the reviewer that performing SFT specifically on jailbreak prompts would likely improve ASR against those specific attacks. However, we would like to highlight the experimental context of our claims: to rigorously assess generalization, both SFT and Unlearning methods were trained only on raw harmful queries, without exposure to any jailbreak templates during training. And our conclusions are built on this setting. **We have added clarifications to the context of our conclusion in the revised paper.**
>
> For this generalization setting, our results in Table 1 directly support the claim that unlearning is essential. Under strong adaptive attacks (e.g., GCG, TAP), standard SFT models suffer from high Attack Success Rates (ASR) (e.g., 69.0%, 71.0%), indicating that attackers can easily bypass SFT barriers via optimization. In contrast, Safe Unlearning maintains extremely low ASR (e.g., 5.0%, 3.0%). This significant performance gap demonstrates that SFT does not generalize well to adaptive adversarial optimization, whereas unlearning provides a robust defense layer that is indeed "essential" for resisting sophisticated attacks, justifying our claims in the paper.
>
> > **[Q1]**: I was a bit surprised about the main first set of results just showing how well unlearning works against attacks. I thought the main point of the paper was more about generalization? So how come? Can you explain the choice?
>
> **[R8]**: We included these results to establish a complete evaluation baseline. Before analyzing generalization (the Ripple Effect), it is crucial to verify that the defense works on the training distribution (ID) without compromising General Performance or increasing Over-Refusal rates. Furthermore, the ID results in Table 1 serve as a necessary control group to contrast with the OOD results in Table 2, allowing us to quantify the extent of the generalization.

---

> ### Author Response · Authors · 2025-11-27
> **Looking forward to hear from you**
>
> Dear reviewer uhfA,
>
> Thank you very much for the time and effort you have devoted to reviewing our manuscript. We greatly appreciate your willingness to advocate for acceptance once the raised concerns are addressed. Your feedback is invaluable to us, and we look forward to hearing your thoughts.
>
> Could you kindly take a moment to review our responses and let us know whether they satisfactorily address your concerns? If anything remains unclear, we would be glad to clarify further.
>
> Best regards,
>
> Authors

---

### Official Review · Reviewer_AUfc · 2025-10-31

**Soundness:** 4
**Presentation:** 3
**Contribution:** 3
**Rating:** 4
**Confidence:** 4

**Summary:**

The paper studies unlearning as a defense against jailbreaks and reports a ``ripple effect'': after unlearning harmful responses for a subset of topics (e.g., theft), LLMs also reduce harmfulness on unseen topics (e.g., bomb-making). On Llama-3.1-8B-Instruct and Mistral-7B-Instruct, across different attacks (GCG, AutoDAN, PAIR, SAA, TAP), unlearning methods (SFT, DPO, NPO, RMU) decrease the Attack Success Rate on unseen harmful questions from 70% to less than 10% with only 100 training samples. The paper also reveals that the strong generalization ability of unlearning may stem from the intrinsic relatedness among harmful responses.

**Strengths:**

1. The paper demonstrates the ripple effect in unlearning, where the model can also forget the unseen, harmful knowledge during unlearning. This insight is both novel and valuable for designing more generalizable jailbreak defenses.

2. The experimental coverage is strong: the paper evaluates multiple unlearning methods and provides a thoughtful analysis of why OOD harmful knowledge is suppressed.

**Weaknesses:**

1. While the ripple effect is interesting, the paper largely stops at verifying this phenomenon. I am considering whether the contribution is enough. Basically, a further concrete method or optimization exploiting the ripple effect to yield a stronger or tunable jailbreak defense would greatly improve the contribution of the paper.

2. The key point -- harmful prompts or responses cluster in the latent space and thus enable OOD harmful forgetting -- is demonstrated primarily on a single benchmark (HarmBench). Without considering across multiple harmful datasets(e.g., [BeaverTails](https://huggingface.co/datasets/PKU-Alignment/BeaverTails) and others), it is unclear whether the effect is benchmark-specific. The experimental breadth on jailbreaks and unlearning is good, but the phenomenon claim needs multi-dataset evidence.

**Questions:**

1. Why does the paper only rely on Llama-3-8B-Lexi-Uncensored to generate harmful responses instead of using the datasets that already contain harmful responses, like [Antropic/hh-rlhf](https://huggingface.co/datasets/Anthropic/hh-rlhf/viewer/default/train?row=0&views%5B%5D=train)? Or using different uncensored models to generate harmful responses may be a good choice. Do you think that only relying on one model could result in harmful responses from the same distribution?

2. As the paper claims that **OOD harmful responses often share common harmful or general expressions with those encountered during training**, 1) does that mean if the attacker changes to an uncommon expression targeting the same harmful behavior, the ripple effect would go away? 2) If so, should the phenomenon of the ripple effect occur if and only if the training and testing have a similar expression? 3) If they have similar expressions, why are the testing harmful questions OOD?

---

> ### Author Response · Authors · 2025-11-24
> **Response by Authors (1/2)**
>
> We sincerely thank the reviewer for the thoughtful and encouraging feedback. We are grateful for the reviewer’s recognition of the novelty and value of our central insight—the ripple effect in unlearning, where removing a specific harmful capability can also suppress broader, unseen harmful knowledge. We are also encouraged by the reviewer’s positive assessment of our extensive empirical coverage, including evaluations across multiple unlearning methods and our analysis explaining why out-of-distribution harmful behaviors are mitigated. Below, we respond to the reviewer’s questions and concerns in detail.
>
> > **[W1]**: While the ripple effect is interesting, the paper largely stops at verifying this phenomenon. I am considering whether the contribution is enough. Basically, a further concrete method or optimization exploiting the ripple effect to yield a stronger or tunable jailbreak defense would greatly improve the contribution of the paper.
>
> **[R1]**: For the contribution question, we would like to emphysis that our paper is the first to systematically uncover that harmful behaviors form stable, coherent clusters in representation space, and that unlearning reshapes these shared subspaces, leading to predictable cross-sample suppression. This goes beyond reporting observations: we introduce new analyses, controlled perturbation experiments, and mechanistic explanations to reveal the structural conditions under which unlearning unexpectedly generalizes. Importantly, novelty and contribution in ICLR does not require a new algorithm. A significant body of prior ICLR papers are phenomenon-discovery, measurement, or mechanistic-analysis works, and many have been accepted. Our contribution is of this type:
> - we identify a previously unknown behavior of existing unlearning methods,
> - we provide representational and behavioral evidence explaining it, and
> - we establish new empirical regularities that future algorithms should take into account.
>
> In summary, our novelty lies not in introducing another algorithm, but in uncovering a fundamental and previously unrecognized mechanism in unlearning. We believe this is also exactly the type of scientific contribution appropriate for ICLR.
>
> > **[W2]**: The key point -- harmful prompts or responses cluster in the latent space and thus enable OOD harmful forgetting -- is demonstrated primarily on a single benchmark (HarmBench). Without considering across multiple harmful datasets(e.g., BeaverTails and others), it is unclear whether the effect is benchmark-specific. The experimental breadth on jailbreaks and unlearning is good, but the phenomenon claim needs multi-dataset evidence.
>
> **[R2]**:  We appreciate this question. Evaluating our method on multiple datasets to further verify OOD generalization is indeed a valuable suggestion.
> In our main text, we currently analyze OOD generalization based on HarmBench, and we have also included evaluations on AIR-Bench in Appendix D. Given that BeaverTails is not well-suited for jailbreak evaluation scenarios, we have adopted the StrongREJECT [1] dataset for testing.
> The results on Llama-3.1 and Mistral-v0.3 are as follows:
>
> **Results on Llama-3.1**:
> | **Method** | Raw | **Manual** | **Prefill** | **PAIR** | **PPL** |
> |--|--|-|-|--|--|
> | **No Defense** | 13.33 | 52.67 | 26.67 | 61.67 | 3.76 |
> | **SFT** | 5.00 | 43.33 | 15.00 | 56.67 | 4.11 |
> | **DPO1 (Large $\beta$)** | **0.00** | 21.00 | 0.00 | 25.00 | 5.35 |
> | **DPO2 (Small $\beta$)** | **0.00** | 0.17 | 3.33 | **1.67** | 3.26e8 |
> | **RMU** | 6.67 | 3.00 | 6.67 | 6.67 | 1.14e4 |
> | **Circuit Breaker** | 10.00 | 24.83 | 20.00 | 50.00 | 2.01e6 |
> | **Safe Unlearning** | **0.00** | **0.00** | **0.00** | 6.67 | 1.62e10 |
>
> **Results on Mistral-v0.3**:
> | **Method**  | **R****a****w** | **Manual** | **Prefill** | **PAIR**  | **PPL** |
> | -- | --- | - | --| - | - |
> | **No Defense**   | 50.00 | 75.00      | 65.00       | 61.67     | 2.86    |
> | **SFT**   | 30.00 | 34.67      | 20.00       | 61.67     | 2.85    |
> | **DPO_1 (Large $\beta$)** | 1.67 | 13.50      | **0.00**    | 26.67     | 14.38   |
> | **DPO_2 (Small $\beta$)** | 1.67   | **0.67**   | 0.00        | **15.00** | 1.77e7  |
> | **RMU**   | 20.00 | 40.33  | 13.33       | 35.00     | 2.60e3  |
> | **Circuit Breaker**    | **0.00**        | 9.83 | 1.67        | 21.67     | 1.32e4  |
> | **Safe Unlearning**   | 3.33            | 2.33       | 1.67        | 26.67     | 1.28e6  |
>
> We can observe that these results are generally consistent with the HarmBench OOD tests reported in the paper. Unlearning-based methods demonstrate strong defense efficacy across various attack methods. The high PPL on harmful responses also indicate similar effective removal of harmful knowledge (e.g., Safe Unlearning reaches 1.28e6 on Llama-3.1 and 1.62e10 on Mistral-v0.3, and DPO_2 reaches 1.77e7 and 3.26e8 respectively).
>
> [1] Souly, Alexandra, et al. "A strongreject for empty jailbreaks." Advances in Neural Information Processing Systems 37 (2024): 125416-125440.

---

> ### Author Response · Authors · 2025-11-24
> **Response by Authors (2/2)**
>
> > **[Q1]**: Why does the paper only rely on Llama-3-8B-Lexi-Uncensored to generate harmful responses instead of using the datasets that already contain harmful responses, like Antropic/hh-rlhf? Or using different uncensored models to generate harmful responses may be a good choice. Do you think that only relying on one model could result in harmful responses from the same distribution?
>
> **[R3]**: We thank the reviewer for raising this insightful point regarding the source and distribution of harmful responses. We did't use Antropic/hh-rlhf as its queries differ substantially from those in standard jailbreak benchmarks like HarmBench. Our approach is grounded in the premise that harmful and benign knowledge form topologically distinct clusters in the representation space; we hypothesized that this separation is structural and largely independent of the specific source model.
>
> To empirically validate this and address the concern about distribution bias, we have conducted additional experiments using Qwen3-8B-Uncensored and Gemma3-4B-Uncensored, alongside the original Llama-3-8B-Lexi-Uncensored. As detailed in the new Appendix C (Discussion on Harmful Response Sampling), we performed a t-SNE analysis on harmful responses sampled from all three models. The results demonstrate a consistent separation pattern that aligns with our main paper’s findings. **This confirms that harmful responses from diverse models cluster consistently within the "harmful" region of the representation space, validating the robustness of our findings.**
>
>
>
> > **[Q2]**: As the paper claims that OOD harmful responses often share common harmful or general expressions with those encountered during training, 1) does that mean if the attacker changes to an uncommon expression targeting the same harmful behavior, the ripple effect would go away? 2) If so, should the phenomenon of the ripple effect occur if and only if the training and testing have a similar expression? 3) If they have similar expressions, why are the testing harmful questions OOD?
>
> **[R4]**: We would like to clarify that the ripple effect is driven by the intrinsic relatedness of harmful outputs, rather than the similarity of input expressions.
>
> Regarding your first two questions: Changing to uncommon input expressions does not eliminate the ripple effect. While attackers may manipulate input distributions (e.g., using rare phrasing) to bypass filters, the elicited harmful responses inevitably converge on shared patterns and cluster in the model's latent space. Since our method suppresses these shared output representations, the defense generalizes regardless of input similarity.
>
> Regarding the definition of "OOD": We classify testing questions as OOD based on distinct topics and intents (e.g., training on "Theft" vs. testing on "Bomb-making"). While these tasks are semantically distinct (requiring different specific knowledge), their responses often share structural commonalities—such as affirmative prefixes and clustered representations. It is these shared output characteristics, not input similarities, that enable the ripple effect to generalize across different harmful topics.

---

> > ### Comment · Reviewer_AUfc · 2025-11-27
> >
> > Thank you for the comprehensive response and the additional experiments. My concerns about OOD unlearning generalization on multiple datasets and OOD harmful response are addressed. However, I have the same concern with the reviewer nVCM and skqa that the contribution of this work does not support it for the main conference publication. I decide to keep my score unchanged.

---

### Official Review · Reviewer_n4PZ · 2025-11-01

**Soundness:** 3
**Presentation:** 3
**Contribution:** 3
**Rating:** 8
**Confidence:** 2

**Summary:**

This paper focuses on the unlearning-based defenses against jailbreaking attacks and introduce the ripple effect that could help improve the models' robustness against unseen adversarial prompts. The authors conducted over 100 experiments across various modes, attack strategies and defense, which validates the effectiveness of the unlearning-based defense. This paper also compare the unlearning-based defense (i.e., unlearning harmful response) to merely learning harmless response.

**Strengths:**

1. The idea of unlearning-based defense is interesting. Intuitively, learning harmless response would unavoidably hurt the models' helpfulness while unlearning-based defenses do not suffer from this drawback.
2. The experimental results are convincing. Results in Section 4 exhibit clear improvement in the robustness of the models (measured by ASR).

**Weaknesses:**

1. The related works on unlearning-based defenses (or simply machine unlearning) is not detailedly discussed.

**Questions:**

1. Does the proposed method have any known negative side effect?

---

> ### Author Response · Authors · 2025-11-24
> **Response by Authors**
>
> We sincerely thank the reviewer for the thoughtful and constructive feedback. We are encouraged by the reviewer’s recognition of the novelty and intuition behind unlearning-based defenses—specifically, that unlike learning-based defenses, unlearning-based methods avoid the inherent trade-off between safety and helpfulness. We also greatly appreciate the reviewer’s positive assessment of the empirical results, especially the clear improvements in robustness (ASR reduction) demonstrated across our experiments in Section 4. Below, we address the reviewer’s questions and concerns in detail.
>
> > **[W1]**: The related works on unlearning-based defenses (or simply machine unlearning) is not detailedly discussed.
>
> **[R1]**: We thank the reviewer for the helpful suggestion. In the revised version, we have expanded the discussion of unlearning-based defenses.
>
> > **[Q1]**: Does the proposed method have any known negative side effect?
>
> **[R2]**: We have not observed clear negative side effects. As reported in Section 5.2, the influence on related benign knowledge is limited and remains within an acceptable range. A more exhaustive study of potential edge cases and long-term effects is an important direction that we plan to pursue in future work.

---

### Author Response · Authors · 2025-11-27
**Summary of Rebuttal**

Dear AC,

Given the unusually high load on ACs after the emergent incident, we believe it is helpful to concisely summarize the rebuttal progress to facilitate a quick and holistic understanding of the current status. We also sincerely thank you in advance for the time and effort you will dedicate to evaluating our work.

We summarize the rebuttal by reviewer:
## Reviewer AUfc, nVCM and skqa
(Note: reviewer skqa **has raised the score from 2 to 4**, as reflected in our discussion.)

Currently, reviewers AUfc, nVCM and skqa have provided feedback. Based on our discussions and their comments, **all detailed concerns have been addressed**, and the **sole remaining question** is whether the contribution of this paper is enough. We therefore wish to further clarify and emphasize the contributions:
- We **identify a previously unknown generalization behavior (i.e., the ripple effect) of unlearning methods** through over **100** runs spanning diverse models, attacks, and defenses. We also propose Safe Unlearning based on NPO, which is more suitable for safety scenarios, although we don't view this as the primary contribution.
- We **formally define** what the ripple effect is, and **throughly discuss the significant differences from prior consensus on unlearning and safety generalization**.
- We explain **what are unlearned** by visualizing the token probability distributions after unlearning, and provide **representational and behavioral evidence explaining the ripple effect** (i.e., representation clustering and shared/similar expressions for different harmful responses).
- We thoroughly discuss the **generalization boundaries** of the ripple effect, by carefully controlling the train set size and topic shift. Based on this, we **provide a practical guidance on how to perform unlearning effectively and efficiently**. We also evaluate its **impact on related benign knowledge**.
- We establish new empirical regularities that **future algorithms should take into account**.

Moreover, reviewers acknowledge that the identified ripple effect is novel (AUfc), valuable (AUfc), interesting (uhfA, skqa), good (nVCM) or convincing (n4PZ). We therefore believe that the insights provided in this work are of clear relevance to the community, and that the above contributions justify publication at the main conference.

Reviewers AUfc, nVCM, and skqa mainly think that observation and analysis alone may not be sufficient, and that proposing new algorithms or theoretical proofs would be necessary for a main-conference contribution. However, many influential papers at ICLR and NeurIPS are centered on phenomenon discovery, measurement, or mechanistic analysis, without introducing new algorithms, and they have been accepted as significant scientific contributions [1,2,3].

Therefore, we believe that the claim of limited contribution due to being observational/analytical only is overly general and not an adequate ground for rejection—indeed, it could be applied to any observational and analytical study, making it highly subjective. We sincerely hope the AC would carefully consider the evaluation of this work in light of the above points.

## Reviewer n4PZ
The reviewer gave a highly positive score. We have addressed their concerns by adding related work discussion and clarifying the noted side effect.

## Reviewer uhfA
Although the reviewer initially gave a score of 4 and did not engage in the rebuttal, the initial review explicitly states a willingness to advocate for acceptance if concerns are addressed. We have made substantial efforts to do so:
- **Writing improvements**. We improved the Introduction and added a clearer explanation of Safe Unlearning in the main paper.
- **Enhanced related work**. We added citations and expanded discussions to related works in the revised paper.
- **Clarifications**. We clarified the distinction from prior safety generalization, noted that strong adaptive attacks are already included in our experiments, added context to one claim in the revised paper, and explained an experiment design choice.

We believe our responses sufficiently address the reviewer’s concerns.

## Conclusion

We believe we have fully addressed most reviewers’ concerns. The only unresolved point regarding the adequacy of an observational/analytical contribution, is overly general and not persuasive in light of precedent. We sincerely expect the AC to consider our arguments above when making the final assessment.
***
**References**

[1] Yue, Yang, et al. "Does reinforcement learning really incentivize reasoning capacity in llms beyond the base model?." NeurIPS 2025 Best  Paper Runner Up.

[2] Zhou, Zhenhong, et al. "On the Role of Attention Heads in Large Language Model Safety." ICLR 2025.

[3] Bianchi, Federico, et al. "Safety-Tuned LLaMAs: Lessons From Improving the Safety of Large Language Models that Follow Instructions." ICLR 2024.

---

### Meta-Review · Area_Chair_bCFR · 2025-12-10

**Summary:**

This paper investigates and shows unlearning generalizes across topics, and identify a novel ripple effect of unlearning, wherein LLMs can implicitly unlearn harmful knowledge that was not explicitly introduced during the unlearning phase. There also discuss the generalization boundary of the observed ripple effect.

**Reviewer Concerns:**

While the ripple effect is interesting, the paper largely stops at verifying this phenomenon. The reviewer is concerning the contribution of this paper. Reviewer think that a further concrete method or optimization exploiting the ripple effect to yield a stronger or tunable jailbreak defense would greatly improve the contribution of the paper.

Without considering across multiple harmful datasets(e.g., BeaverTails and others), it is unclear whether the effect is benchmark-specific.

**Reviewer Scores:**

Most reviewers have the same concern that the contribution of this work does not support it for the main conference publication. They have decided to keep their score unchanged.

---

### Decision · Program_Chairs · 2026-01-26

Reject